# Experiences and challenges in accessing hospitalization in a government-funded health insurance scheme: Evidence from early implementation of Pradhan Mantri Jan Aarogya Yojana (PM-JAY) in India

**Mayur Trivedi**[1‡]*, **Anurag Saxena**[1‡], **Zubin Shroff**[2], **Manas Sharma**[1]

**1** Indian Institute of Public Health, Gandhinagar, Gujarat, India, **2** World Health Organization, Geneva, Switzerland

‡ MT and AS are co-first authors on this work.
* mtrivedi@iiphg.org

**Data Availability Statement:** This data contained in this study were made available to the research

## Abstract

### Introduction

Government-sponsored health insurance schemes can play an important role in improving the reach of healthcare services. Launched in 2018 in India, Pradhan Mantri Jan Aarogya Yojana (PM-JAY) is one of the world's largest government-sponsored health insurance schemes. The objective of this study is to understand beneficiaries' experience of availing healthcare services at the empaneled hospitals in PM-JAY. This study examines the responsiveness of PM-JAY by measuring the prompt attention in service delivery, and access to information by the beneficiaries; financial burden experienced by the beneficiaries; and beneficiary's satisfaction with the experience of hospitalization under PMJAY and its determinants.

### Methods

The study was conducted during March-August 2019. Data were obtained through a survey conducted with 200 PM-JAY beneficiaries (or their caregivers) in the Indian states of Gujarat and Madhya Pradesh. The study population comprised of patients who received healthcare services at 14 study hospitals in April 2019. Prompt attention was measured in the form of a) effectiveness of helpdesk, and b) time taken at different stages of hospitalization and discharge events. Access to information by the beneficiaries was measured using the frequency and purpose of text messages and phone calls from the scheme authorities to the beneficiaries. The financial burden was measured in terms of the incidence and magnitude of out-of-pocket payments made by the beneficiaries separate from the cashless payment provided to hospitals by PMJAY. Beneficiaries' satisfaction was measured on a five-point Likert scale.

team by the National Health Authority, India for exclusive use in this study. The data belong to the National Health Authority. For more information please contact Dr Ruchira Agrawal , Consultant M&R, National Health Authority at Mel.cons1@nha.gov.in."

**Funding:** The research was made possible through the financial support provided by the World Health Organization. This included support from the Alliance for Health Policy and Systems Research, WHO Geneva as well as from the WHO Country Office, India under grant numbers 68345 (WHO, Geneva) and 67378 (WHO, India). Support was provided under the research program titled- Health Systems Research for PM-JAY: Improving hospital-based processes for effective implementation. The grant was received by Prof. Dileep Mavalankar, Prof. Mayur Trivedi, and Prof. Anurag Saxena of the Indian Institute of Public Health Gandhinagar. The grant period was 4 March to 3 September 2019. The URL of the funder is www.who.int. Dr. Zubin Shroff of WHO provided guidance in strengthening the methodology and finalizing the instruments, and reviewed the manuscript critically.

**Competing interests:** The authors have declared that no competing interests exist.

## Results

Socio-economically weaker sections of the society are availing healthcare services under PM-JAY. In Gujarat, the majority of the beneficiaries were made aware of the scheme by the government official channels. In Madhya Pradesh, the majority of the beneficiaries got to know about the scheme from informal sources. For most of the elements of prompt attention, access to information, and beneficiaries' satisfaction, hospitals in Gujarat performed significantly better than the hospitals in Madhya Pradesh. Similarly, for most of the elements of prompt attention, access to information, and beneficiaries' satisfaction, public hospitals performed significantly better than private hospitals. Incidence and magnitude of out-of-pocket payments were significantly higher in Madhya Pradesh as compared to Gujarat, and in private hospitals as compared to the public hospitals.

## Conclusion

There is a need to focus on Information, Education, and Communication (IEC) activities for PM-JAY, especially in Madhya Pradesh. Capacity-building efforts need to be prioritized for private hospitals as compared to public hospitals, and for Madhya Pradesh as compared to Gujarat. There is a need to focus on enhancing the responsiveness of the scheme, and timely exchange of information with beneficiaries. There is also an urgent need for measures aimed at reducing the out-of-pocket payments made by the beneficiaries.

## Introduction

An important goal of the health system of any country is to improve the health outcomes of its citizens. However, even a health system that successfully attains desired health outcomes can be deemed unsuccessful if it fails to satisfy its users. Users can disapprove of a health system if treatment costs are high or if the system does not provide a timely response to their felt needs. Hence, while working to preserve, promote, and improve the population's health, health systems must also strive to provide financial protection and timely response to the expectations of the population [1].

Health system responsiveness is defined as "the ability of the health system to meet the population's legitimate expectations regarding their interaction with the health system, apart from expectations for improvements in health or wealth" [2]. Responsiveness includes both physical and affective support during treatment. It incorporates elements of respect for and orientation to the rights of clients including confidentiality, prompt attention, choice of providers, and the quality of amenities. The flow and clarity of information between the health system and its patients also form an important element of responsiveness [3]. Information asymmetries between patients and providers are typical in healthcare—however, a responsive health system not only aims to keep patients informed about the course of treatment or other actions recommended by the healthcare providers but also has a mechanism to solicit patients' feedback [4].

Government-sponsored health insurance schemes (GSHIS) can play an important role in developing countries. For developing countries, GSHISs have been advocated as a means for governments to fulfill their responsibilities to citizens, including their commitments to move towards the Universal Health Coverage (UHC) as envisioned under target 3.8 of the United Nation's Sustainable Development Goals (SDGs) [5]. The element of responsiveness has been said to be especially essential for GSHIS that seeks to enhance the provision of services to the citizens through strategic purchasing of healthcare services [2, 6].

India is one of the countries that has been rapidly expanding health insurance coverage. In September 2018, the Government of India launched Pradhan Mantri Jan Aarogya Yojana (PM-JAY) to provide insurance coverage to approximately 500 million poor and vulnerable beneficiaries forming the bottom 40% of the Indian population [7]. The scheme provides coverage of up to INR 500,000 (USD 6800) per family per year for all secondary and most of the tertiary care procedures of surgery, medical, and daycare treatments at public and empaneled private hospitals [8]. The scheme is rolled out in active partnership with the state governments wherein state governments are implementing the scheme through a trust or an insurance company. This trust or an insurance company empanels the hospitals, approves the pre-authorization request, and makes payment to the hospitals.

To facilitate the implementation of PM-JAY, National Health Authority (central coordinating agency for PM-JAY) has provided guidelines for each step of the hospitalization process, namely, patient's registration, selection of treatment package, preauthorization and hospitalization, discharge, claim reimbursement to the hospital, and exchange of information by the scheme authorities with the beneficiaries [9]. To support beneficiaries' engagement with PM-JAY, the guidelines state that all empaneled hospitals must place a dedicated PM-JAY helpdesk at a prominent location in the hospital. The primary purpose of this helpdesk is to act as a 'one-stop' point for the information and help needed by the PM-JAY beneficiaries. This helpdesk is to be managed by an Ayushman Mitra (AM), a hospital representative who manages operations and liaises between patients, doctors, hospital administration, and the scheme managers. The guidelines state that the AM should be able to use a diagnosis sheet provided by the doctor to select and block the treatment package(s) in the PM-JAY IT (Information Technology) system. Once AM selects and blocks the package in the PM-JAY IT system, the request is submitted to the insurer for pre-authorization. According to guidelines, the insurer must either approve or reject this pre-authorization request within six hours of receiving the request. Once the pre-authorization request is approved by the insurer, patients are hospitalized and are provided the treatment. Being a cashless scheme, patients are not supposed to make any payments during any stage of hospitalization. PM-JAY guidelines also require scheme authorities to contact patients through text messages (Short Message Service) and telephonic calls to keep them informed and take their feedback on the hospitalization process [10]. As per the guidelines, some procedures, deemed prone to fraud and abuse, are reserved for public hospitals only. Beneficiaries must bear the expenses for these procedures if treatment is availed in a private hospital [11].

PM-JAY is one of the world's largest government-sponsored health insurance schemes and is aimed at providing health insurance coverage to a large number of economically poor Indian citizens. This population, in India, has previously been considered a passive recipient of healthcare services, often has to bear catastrophic healthcare expenditure, and neither public nor private sector has been responsive to them. Within PM-JAY, though guidelines for the processes to be followed in the scheme are in place, however, there is a lack of information on beneficiaries' experiences of navigating hospitalization processes, accessing information, and the financial burden incurred by them. In this context, this research aims to understand the experiences of beneficiaries who availed healthcare services under PM-JAY at the empaneled public and private hospitals. Specifically, this study aims to understand beneficiaries' experiences of identification and registration in PM-JAY; navigating hospitalization processes related to medical package selection, preauthorization, discharge, and services received; information exchange during beneficiary verification, before hospitalization, during discharge, and after hospitalization; any additional payments made by the beneficiaries to the hospital separate from the cashless payments by PM-JAY, and beneficiaries' satisfaction with the experience of hospitalization under PMJAY. The study is looking at the element of responsiveness in

PM-JAY by measuring the prompt attention in service delivery by the providers/scheme authorities, and access to information by the beneficiaries. This study is looking at the financial burden in terms of incidence and magnitude of out-of-pocket payments made by the beneficiaries to the hospital. Along with it, this study also attempts to understand beneficiary's satisfaction with the experience of hospitalization under PMJAY and its determinants.

Much of the published discourse around health system responsiveness on the experiences and expectations of patients is limited to developed country settings. Only a few studies have explored it in the context of developing countries [12–14]. PM-JAY is a recent initiative and there is limited literature on it. In the existing literature, a need to study beneficiaries' perspectives on PM-JAY processes has also been stated [15]. By attempting to understand responsiveness, financial burden, and patient satisfaction and its determinants, this study seeks to enrich the existing literature and provides useful insights to the policymakers and program managers to strengthen the scheme.

## Methodology

This study was conducted between March and August 2019. In consultation with the National Health Authority (NHA), Gujarat and Madhya Pradesh states in India were selected as the study locations. Gujarat is an economically developed state with a higher annual per capita Gross State Domestic Product, as compared to Madhya Pradesh. In the recent past, Gujarat had a high level of per capita health expenditure as compared to Madhya Pradesh, and the total and out-pocket expenditure per episode of hospitalization was higher in Madhya Pradesh. (S1 Table). Prior to PM-JAY, Gujarat has an experience of implementing state government funded GSHIS (Mukhyamantri Amrutam scheme) whereas Madhya Pradesh has no experience of implementing entitlement-based GSHIS. Study sites included seven hospitals in each state. These hospitals were selected by the respective State Health Agency (SHA) after ensuring representation of a) public and private hospitals, and b) multi-specialty and super-specialty hospitals (SHA is a state-level nodal agency for the implementation of PM-JAY). As shown in Table 1, the fourteen hospitals included in the study comprised nine multi-specialty and five super-specialty hospitals. Eight of the hospitals were private and the rest were public hospitals.

The study population was drawn from patients who were hospitalized in April 2019. Sample selection was done using the list of all claims from study hospitals, as provided by the SHAs in the two states. Each claims list was converted into a patient list by retaining one claim per beneficiary. Next, to ensure representation of most and least popular services at each hospital, the most and least popular specialties were selected. For each hospital, specialty clusters were arranged from those having the highest number of patients to those having the lowest. Those clusters that together contained the top 10% of patients were identified as the 'most popular cluster' for that hospital. Similarly, those clusters that together contained the bottom 10% of patients were identified as the 'least popular cluster' for that hospital. The resulting list of patients in the 'top—bottom' clusters was then used to generate a sample of 100 beneficiaries

**Table 1. Study hospitals and surveyed beneficiaries, by state.**

| Hospital type | Ownership type | Gujarat | | Madhya Pradesh | |
|---|---|---|---|---|---|
| | | Number of hospitals | Sample of beneficiaries | Number of hospitals | Sample of beneficiaries |
| Multi-specialty | Private | 2 | 35 | 2 | 29 |
| | Public | 2 | 21 | 3 | 57 |
| Super-specialty | Private | 2 | 21 | 2 | 14 |
| | Public | 1 | 23 | - | - |
| | **Total** | **7** | **100** | **7** | **100** |

in each state through the probability proportional to size (PPS) method using the hospital as the sampling unit. Thus, a higher proportion of the sample was drawn from the hospital with the most claims and the smallest sample was derived from the hospital with the fewest claims. The use of a multi-stage sampling method with the use of PPS ensured that the selection of respondents was not biased either in favor or against any type of specialty and hospital [16]. The final distribution of patients included in the study across the different types of hospitals is shown in Table 1. The sampled patients were from six districts in Gujarat and five districts in Madhya Pradesh. Among the 100 patients in Gujarat, 39 were only enrolled in the Mukhya-mantri Amrutam critical illness coverage scheme that existed at the time, not the full PM-JAY scheme (which has since subsumed Mukhyamantri Amrutam).

Data for this study were obtained through a household survey of the sampled beneficiaries. A structured questionnaire was developed to solicit details of the patient's hospitalization experience keeping in consideration PM-JAY guidelines. The draft questionnaire was reviewed and approved by technical experts at the World Health Organization (WHO) and the NHA. Before data collection commenced, the revised instrument was piloted for accuracy and validation through a discussion with a panel of in-house experts and a field test. The survey questionnaire is available as supplementary material to this paper (S1 File). The data collection was done by a team of six investigators in May and June 2019. Before going to the field, these investigators underwent a one-day training specifically designed for the data collection targeted under this research. The questions were asked in the local language i.e. Gujarati for Gujarat and Hindi for Madhya Pradesh. Each survey administration lasted for 30–40 minutes.

The survey was administered to—either the patient or his/her primary caregiver—at their residences. They were contacted telephonically in advance to ascertain their availability for the survey. All participants were provided with a participant information sheet that described the ethical concerns and emphasized the rights of the respondents. After providing a brief description of the study and its objectives, surveyors requested the respondent's participation in the survey, and verbal non-witnessed informed consent was taken. Nine of the initially contacted respondents refused to participate in the survey. These respondents were replaced by another nine participants from the patient list.

The consented respondents were asked to indicate the reasons for and duration of their hospitalization. They were asked in detail about their experiences and any challenges experienced during beneficiary identification and authentication, the experience of hospitalization, and the scope of services received (including medicines). The prompt attention element of responsiveness was measured in the form of a) extent to which help was provided at the help-desk, b) time taken at different stages of hospitalization and discharge process, and c) information exchange between patient and scheme authorities. To ascertain the extent to which help was provided at the helpdesk, respondents were asked closed-ended questions about whether they had received three types of help from the registration desk: 1) information about PM-JAY; 2) help with documents and computerized registration; and, 3) guidance about treatment within the hospital. Respondents were also asked to report how long the pre-authorization request, admission, and discharge processes took. Respondents were also asked to provide details of any message and telephone calls received from the scheme authorities about registration, preauthorization, hospitalization, and discharge. Respondents were asked to rate their experiences of registration, and hospitalization on a five-point Likert scale that ranged from 'highly dissatisfied' to 'highly satisfied'. The financial burden was measured in terms of OOP payment made by the patients to the hospitals. To ascertain the incidence and magnitude of OOP payments, respondents were asked to provide information on any additional payments made to the hospital separate from the cashless payment provided to hospitals by PMJAY. Respondents were also asked to indicate the reason for which the payment was made.

The questionnaire for the household survey was configured using Open Data Kit (ODK) software, a free and open-source tool developed by the University of Washington. The data was collected using hand-held electronic devices. On-site monitoring and cross-verification of data collection were carried out to ascertain the reliability of the data being collected. This was done by the research supervisors through observation and spot-checking.

The collected data was cleaned using MS-Excel software and descriptive statistics were generated. Data was statistically analyzed using SPSS version 20. Non-parametric test (Mann–Whitney U test) was used to compare beneficiaries' responses across states (Gujarat and Madhya Pradesh), hospitals across states (public and private hospitals), and hospitals within a state (public and private hospital in Gujarat, public and private hospital in Madhya Pradesh). The non-parametric test (Mann–Whitney U test) was used due to the non-normal distribution of data (Kolmogorov-Smirnov and Shapiro-Wilks test, $p < 0.05$). A significance level of $P < 0.05$ was used for the Mann–Whitney U test. For each set of comparisons mean, standard deviation, and median of the studied variable was noted. Along with it, Mann–Whitney U test statistics and p-value were also noted. Before conducting the comparisons, extreme outliers were detected using a box-plot and removed from the dataset. Extreme outliers were defined as data points that were more than 3 box-lengths away from the edge of their box in the box-plot. These extreme outliers were less than five percent of the dataset under consideration.

A cumulative odds ordinal logistic regression with proportional odds was conducted to determine the effect of state, hospital type, OOP incidence, beneficiary's caste, beneficiary's location, beneficiary level of education, type of care/services received, help received at PM-JAY registration desk, and the number of days for which the beneficiary was admitted in a hospital (independent variables), on the beneficiary's satisfaction with the experience of hospitalization under PMJAY (dependent variable). The dependent variable beneficiary's satisfaction with the experience of hospitalization under PMJAY was an ordinal variable measured on a five-point scale that ranged from 'highly dissatisfied' to 'highly satisfied'. The independent variables of

- State refers to the state in which the respondent lives (Gujarat, MP)

- Hospital type refers to the type of hospital in which respondent underwent hospitalization (Public, Private)

- OOP made indicates whether respondent made OOP (Yes, No)

- Beneficiary caste indicate the social caste to which respondent belongs (Marginalized and backward castes, General caste)

- Beneficiary location refers to the geographical area where the respondent lives (urban, non-urban)

- Beneficiary level of education indicate beneficiary's level of schooling (illiterate, primary education, secondary education, and graduate and above)

- Type of care/services received indicates the service that was availed by the beneficiary (hospitalization with surgery, hospitalization without surgery, Daycare procedure)

- Help received at the PM-JAY registration desk indicates the extent of help received by the beneficiary at the registration desk (measured in terms of the help received, namely, information about PM-JAY, help with documents and computerized registration, and guidance about treatment within the hospital)

- The number of days for which the beneficiary was admitted to a hospital indicates the beneficiary's length of stay in the hospital

The study received ethical clearance from the institutional ethics committee of the Indian Institute of Public Health Gandhinagar (IIPHG) (TRC-IEC No: 14/2018-19).

## Results

The results of the study are presented in five sections. The first section provides basic sociodemographic information of the survey respondents, as well as an overview of the healthcare services they received during their hospital visit. The remaining sections are organized based on the beneficiaries' interactions with hospitals and scheme authorities during hospitalization process: identification and registration; treatment package selection before hospitalization; discharge and payments; and, information exchange. Results for prompt attention in service delivery, access to information by the beneficiaries, incidence and magnitude of OOP payment, and beneficiary's satisfaction with the experience of hospitalization under PMJAY are discussed in the flow of beneficiaries' interaction with the system.

### Respondents' characteristics and services received

Table 2 presents a summary of the socio-demographic profile of the survey respondents. The respondent's profile highlight that there is a high proportion of respondents having a low level of education, belonging to socially marginalized and backward castes, and few have a salaried job. This indicates that the socio-economically poorer section of the society is availing healthcare services under PM-JAY.

**Table 2. Profile of the surveyed beneficiaries.**

| Particulars | Details | Gujarat (n = 100) | Madhya Pradesh (n = 100) | Both states (%) |
|---|---|---|---|---|
| Type of hospital | Private | 56 | 43 | 49.5 |
| | Public | 44 | 57 | 50.5 |
| Gender | Female | 38 | 53 | 45.5 |
| | Male | 62 | 47 | 54.5 |
| Mean age (years) | | 49.1 (Range 6–79) | 42.2 (Range 2–75) | |
| Religion | Hindu | 89 | 89 | 89 |
| | Muslim | 11 | 11 | 11 |
| Caste | Marginalized and backward castes (Scheduled Tribe, Scheduled Caste, Other Backward Caste) | 74 | 74 | 74 |
| | Other (General caste) | 26 | 25 | 25.5 |
| | Refuse to answer | 0 | 1 | 0.5 |
| Highest level of education attained in the household | Illiterate or no formal education | 39 | 15 | 27 |
| | Primary education (1–8 standard) | 35 | 50 | 42.5 |
| | Secondary education (9–12 standard and diploma etc.) | 23 | 26 | 24.5 |
| | Graduate and above | 3 | 9 | 6 |
| Occupation | Farm Labor | 18 | 4 | 11 |
| | Other labor work in a rural area | 8 | 7 | 7.5 |
| | Labor work in an urban area | 14 | 22 | 18 |
| | Self-employment (agricultural) | 20 | 7 | 13.5 |
| | Self-employment (non-agricultural) | 19 | 38 | 28.5 |
| | Salaried job | 19 | 22 | 20.5 |
| | Other | 2 | 0 | 1 |
| Respondents who received benefits under state-sponsored health coverage before PMJAY | | 75 | 3 | 39 |

In terms of the services received by the beneficiaries, 39% had hospitalization with surgery, 37% reported hospitalization without surgery, and the remaining respondents received day-care treatment as an outpatient without having to remain overnight at the hospital. These proportions varied across states and types of hospitals. Among the patients who utilized services in public hospitals in Gujarat, 61% received day-care procedures. In Madhya Pradesh, 61% of respondents from public hospitals reported receiving hospitalization without surgery. In both states, around half of the respondents who received treatment in private hospitals had surgical procedures.

## Beneficiary identification and registration

The patient's first touchpoint with the PM-JAY scheme is the beneficiary identification and registration process. This involves beneficiaries' awareness of the insurance scheme and experiences of using the e-card issued to them (PM-JAY e-card indicates beneficiaries' registration in the scheme). The number of respondents from Gujarat in this section is 61, as at the time 39 were enrolled in the Mukhyamantri Amrutam scheme, which had not yet been subsumed under PMJAY; these patients did not have a comparable registration experience.

A large proportion (74%) of PM-JAY beneficiaries in Gujarat indicated that they were made aware of the scheme through a letter from the government. Another 11% indicated that they got to know about the scheme through village-level health workers and 7% got to know about the scheme only at the time of the hospitalization. The remaining respondents got to know about the scheme from friends and relatives or through newspapers. None of the respondents from Madhya Pradesh reported receiving a scheme-related letter from the government. Instead, nearly one-third of the beneficiaries (30%) learned about the scheme when they arrived at the hospital, while one-quarter of the respondents got to know about the scheme from their friends and relatives. The remaining sources included newspaper, Internet, etc.

Most (82%) beneficiaries from Gujarat mentioned that the letter from the government indicated their eligibility for entitlements in the scheme. In Madhya Pradesh, around half of the respondents indicated that they checked their eligibility at common service centers, government-authorized one-stop-facility for availing digital services on various public schemes and entitlements. (Fig 1). This proportion was 63% among those using private hospitals in Madhya Pradesh; around one-third of beneficiaries using public hospitals checked their eligibility at the time of admission in the hospital.

As mentioned, survey respondents were asked about the extent of help they received at the PM-JAY helpdesks. Half of the beneficiaries (52%) reported receiving information about PM-JAY, help with documents and computerized registration, and guidance about treatment within the hospital. Respondents from Gujarat reported receiving relatively higher level of help (Mean = 2.1, Std. Deviation = 0.9, Median = 2) as compared to the respondents from MP (Mean = 2.1, Std. Deviation = 0.9, Median = 2)–no statistically significant difference U = 4793, z = -0.5, p = 0.59. Those who availed services from public hospital reported receiving higher level of help (Mean = 2.3, Std. Deviation = 0.8, Median = 3) as compared to those who availed services from private hospital (Mean = 1.9, Std. Deviation = 0.9, Median = 2)–statistically significant difference U = 6040.5, z = 2.7, p = 0.006. Within Gujarat, patients from public hospitals reported receiving higher level of help (Mean = 2.5, Std. Deviation = 0.8, Median = 3) as compared to the patients from private hospitals (Mean = 1.8, Std. Deviation = 0.9, Median = 1.5)–statistically significant difference U = 1692.5, z = 3.5, p = 0.001. In MP, there was little difference between private (Mean = 2, Std. Deviation = 0.8, Median = 2) and public hospitals (Mean = 2.1, Std. Deviation = 0.9, Median = 2)–no statistically significant difference, U = 1279, z = 0.4, p = 0.7. The proportion of respondents who received all three kinds of help

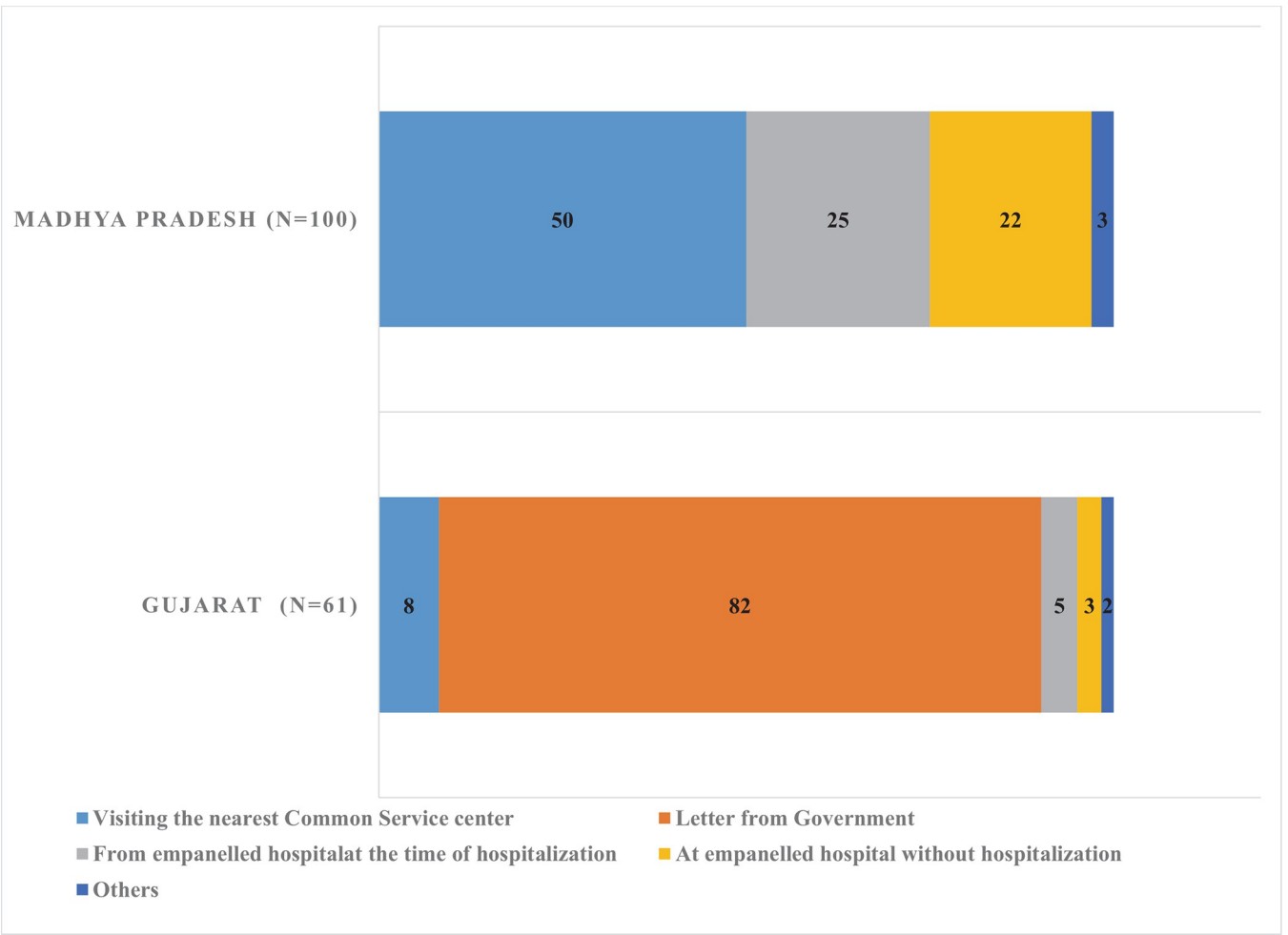

**Fig 1. Proportion of beneficiaries by the source of eligibility checking: A comparison across states (In %).**

was highest in the public hospitals of Gujarat (82%), and lowest in the private hospitals of Madhya Pradesh (35%). Only 7% of respondents indicated that they did not receive any help at the helpdesk—these were respondents who had been at a public hospital in Madhya Pradesh. Fewer than one in ten beneficiaries reported facing problems with registration in the form of long waiting time or payment before registration.

As shown in Fig 2, 81% of respondents felt either 'satisfied' or 'highly satisfied' with the registration process. This was higher in Gujarat (86%) than in Madhya Pradesh (77%), and among beneficiaries from public sector hospitals (83%) than their counterparts from the private sector (77%).

## Treatment package selection and blocking

PMJAY guidelines state that once the patient' eligibility is ascertained, the AM should be able to block the benefit package(s) using PMJAY IT system. The mean waiting time reported by the patients for 'admission and pre-authorization request'–i.e. the time that the AM took in preparing and submitting the request—was 32 minutes in Gujarat (Std Deviation = 25.7, Median = 30) and 75 minutes in Madhya Pradesh (Std Deviation = 104.1, Median = 30)–Statistically significant difference U = 4373.5, z = 4.1, p < 0.005. In Gujarat, the mean reported time

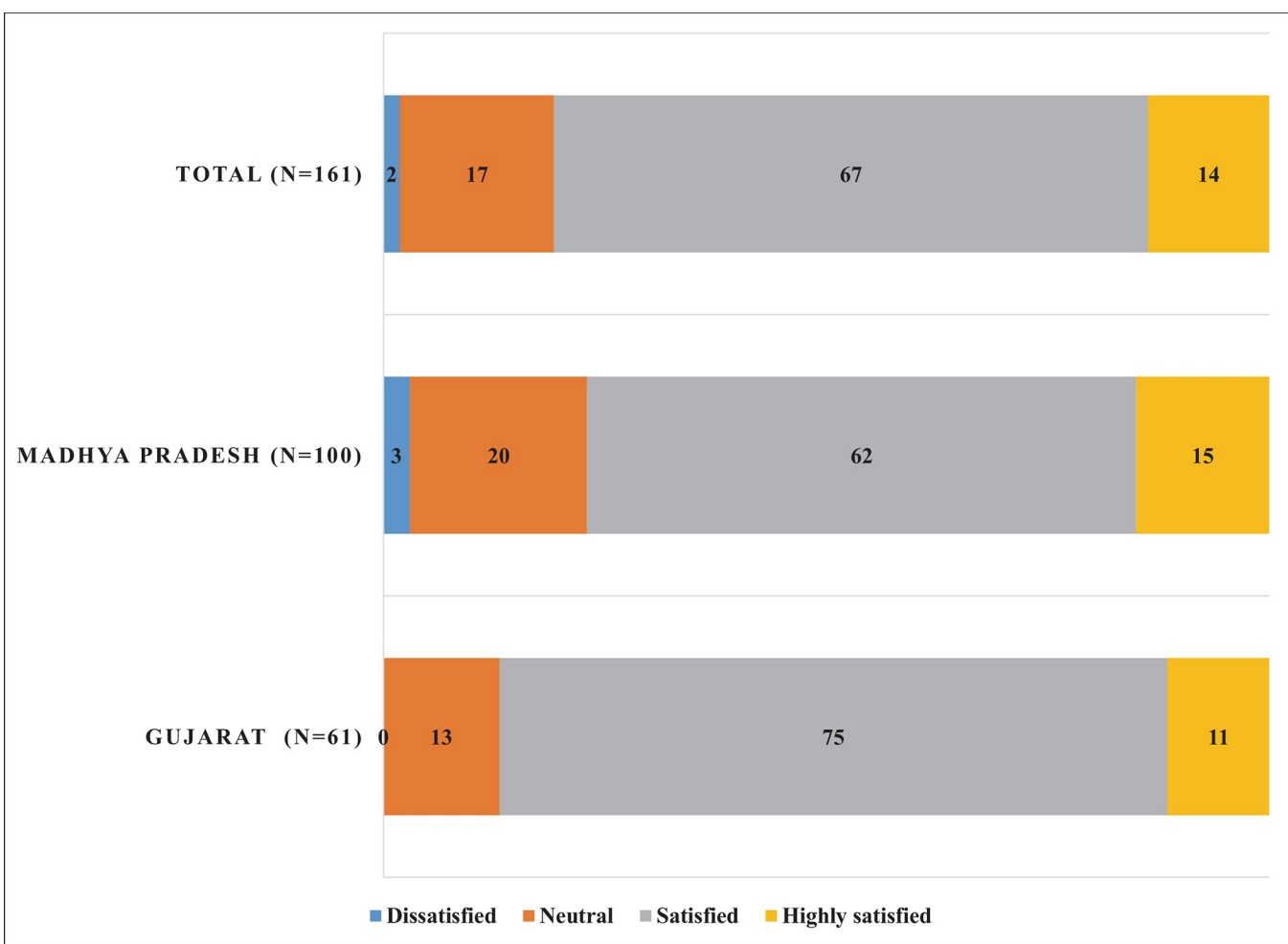

**Fig 2. Proportion of beneficiaries by their level of satisfaction with registration: A comparison across states (In %).**

for 'admission and pre-authorization request' for public hospitals was 29 minutes (Std Deviation = 25.5, Median = 30) and for private hospitals it was 34 minutes (Std Deviation = 25.9, Median = 30)–no statistically significant difference U = 639.5, z = -1.1, p = 0.3. In Madhya Pradesh, the mean reported time was 48 minutes for the patients at public hospitals (Std. Deviation = 51, Median = 30) and 120 minutes for those who were hospitalized in private hospitals (Std. Deviation = 145.6, Median = 50)—statistically significant difference U = 517.5, z = -2.9, p = 0.004. The self-reported mean time for 'admission and pre-authorization request' across the two states for public hospitals was 40 minutes (Std Deviation = 43, Median = 30) and for private hospitals it was 73 minutes (Std. Deviation = 107.5, Median = 30)—statistically significant difference U = 2599, z = -2, p = 0.048.

The average time reported by the beneficiaries for the 'pre-authorization approval' to be received from the scheme authorities was around 150 minutes in Gujarat (Std. Deviation = 625, Median = 30) and 480 minutes in MP (Std. Deviation = 899, Median = 55)–statistically significant difference U = 6123, z = 5, p < 0.005. Across the two states, for public hospitals it was 140 minutes (Std Deviation = 545, Median = 40) and for private hospitals it was 376 minutes (Std. Deviation = 939.4, Median = 60)—no statistically significant difference U = 3769.5, z = -1.5, p = 0.1. In Gujarat, those beneficiaries who availed the service from public hospital reported a

mean time of 184 minutes (Std. Deviation = 680, Median = 30), and those who availed the service from private hospital reported a mean time of 136 minutes (Std. Deviation = 583.3, Median = 30)—no statistically significant difference U = 1313.5, z = 0.7, p = 0.5. In MP, beneficiaries reported a mean time of 104 minutes (Std Deviation = 397.3, Median = 40) for public hospitals and 743 minutes (Std. Deviation = 1230.6, Median = 60) for private hospitals—statistically significant difference U = 382.5, z = -4.664, p < 0.005.

## Discharge and payments

Mean time reported by the beneficiaries to complete the discharge process by the hospitals was 63 minutes (Std. Deviation = 53, Median = 45) in Gujarat and 119 minutes (Std. Deviation = 85, Median = 120) in Madhya Pradesh—statistically significant difference U = 5330.5, z = 5.1, p < 0.005. For Gujarat, mean of the time taken to complete the discharge process was 75 minutes (Std. Deviation = 56.7, Median = 60) for public hospitals, and 52 minutes (Std. Deviation = 47, Median = 30) for private hospitals–statistically significant difference U = 905, z = 2.2, p = 0.03. For Madhya Pradesh, average time taken by public hospitals to complete the discharge process was 106 minutes (Std. Deviation = 76.5, Median = 90) and for private hospitals it was 137 minutes (Std. Deviation = 93.1, Median = 120)–no statistically significant difference U = 923, z = -1.8, p = 0.07. The mean reported time for completion of discharge process across the two states was 94 minutes (Std. Deviation = 71.1, Median = 60) in public hospitals and 96 minutes (Std. Deviation = 79, Median = 60) for private hospitals–no statistically significant difference U = 3950, z = 0.7, p = 0.5.

The average length of stay for hospitalizations, excluding day care procedures, was 4 days (Std. Deviation = 3.2, Median = 3.5 day) in Gujarat and 6 days (Std. Deviation = 2.6, Median = 5 day) in Madhya Pradesh—statistically significant difference, U = 2917.5, z = 2.7, p < 0.005. Average length of stay was 5 days for both surgical treatment (Std. Deviation = 3, Median = 5 days) and non-surgical treatment (Std. Deviation = 2.6, Median = 5 days)—no statistically significant difference, U = 2648, z = 0.7, p = 0.5. The average length of hospitalization was 5 days (Std Deviation = 4.2, Median = 5) in public hospitals and 3 days (Std Deviation = 3, Median = 3) in private hospitals—statistically significant difference, U = 5760.5, z = 2.6, p = 0.009). There was little difference between private (Mean = 5 days, Std. Deviation = 2.5, Median = 5) and public hospitals (Mean = 6 days, Std. Deviation = 2.4, Median = 6) in Madhya Pradesh—no statistically significant difference, U = 1141, z = 1.9, p = 0.05. Private hospitals in Gujarat, however, had noticeably shorter stays (Mean = 3 days, Std Deviation = 2.2, Median = 3) than public hospitals (Mean = 8 days, Std. Deviation = 3.7, Median = 7.5)—statistically significant difference, U = 445.5, z = 4, p < 0.005.

The possibility of having to make additional payments outside of insurance coverage at the time of discharge remained an important concern for the patients. As shown in Table 3, 26% of respondents reported that they made OOP payments either before, during, or after hospitalization. A lower proportion of respondents who reported making OOP payments were from Gujarat (10%) as compared to Madhya Pradesh (42%). A lower proportion (18%) of patients

**Table 3. Incidence of OOP payments during hospitalization (%).**

| Hospital type | Gujarat | Madhya Pradesh | Total |
|---|---|---|---|
| Private | 11 (n = 55) | 63 (n = 43) | 34 (n = 98) |
| Public | 9 (n = 44) | 25 (n = 55) | 18 (n = 99) |
| Mean | 10 (n = 99) | 42 (n = 98) | 26 (n = 197) |

n = number of patients in a quadrant.

from public hospitals made OOP payments as compared to those from private hospitals (34%). The highest incidence of OOP was among patients of private hospitals in Madhya Pradesh.

Most respondents who reported making OOP payments indicated that the payments were for either medicines or diagnostic tests. Patients reported making these payments directly to the pharmacies or laboratories outside the hospitals, as they were told that certain inputs were either not available or not covered under the PM-JAY. In addition, around one-fifth of respondents who made OOP payments, mostly from private hospitals in Madhya Pradesh, indicated that they were asked to make a lump-sum payment directly at the hospital billing counter. These patients reported that they were informed at the helpdesk that the actual costs of their treatments or procedures were higher than the amount the hospital would receive from claim reimbursements from PM-JAY; thus, patients needed to pay the balance.

In terms of the amounts of the OOP payments, there were notable differences between the two states and types of hospitals. The mean OOP payment made by patients in Gujarat was INR 1511 (Std. Deviation = 1620.2, Median = 1000) whereas for Madhya Pradesh it was INR 27,648 (Std. Deviation = 30692.4, Median = 15000)—statistically significant difference $U = 359$, $z = 3.3$, $p = 0.001$. Mean OOP payment made by patients in public hospitals across the two states was INR 1510 (Std. Deviation = 1308.9, Median = 1000) and for private hospitals it was INR 33700 (Std. Deviation = 31342.1, Median = 26000)—statistically significant difference $U = 77$, $z = -4.4$, $p < 0.005$. In Madhya Pradesh, average OOP payment made by patients in public hospitals was INR 1499 (Std. Deviation = 1358.4, Median = 1000) and in private hospitals the mean OOP payment was INR 40996 (Std. Deviation = 30834.8, Median = 34500)—statistically significant difference $U = 10$, $z = -4.9$, $p < 0.005$. In Gujarat, the average amount of OOP payment in public hospitals was INR 1550 and for private hospitals it was INR 1485. Across the two states, the average OOP payment made by the beneficiaries belonging to the marginalized and backward castes was INR 33339 (Std. Deviation = 32157, Median = 25000) and by those from general caste was INR 17608 (Std. Deviation = 26530, Median = 2750)—statistically significant difference $U = 448.5$, $z = 2.3$, $p = 0.022$. No statistically significant difference was found in the OOP payment with respect to the education and occupation of the beneficiaries.

Patients in Gujarat, as well as public hospital patients in Madhya Pradesh who reported lower OOP payments, had to purchase medicine and other supplies from outside the hospital. Most of the respondents from private hospitals of MP who made high OOP payments were hospitalized for either cancer-related procedures or cardiovascular procedures that involved the replacement of a valve or pacemaker. The average OOP payment was around three times higher for those patients who were asked by the hospitals to make a part-payment on top of their insurance coverage (mean: INR 47,840; median: INR. 42,500), when compared with those who reported making OOP for medicines, tests, blood, or other individual items (mean: INR 17,172; median expenses: INR 3,250).

Overall, a majority of survey respondents were either 'satisfied' or 'highly satisfied' with their hospitalization experience. This was higher in Gujarat (82%), compared with Madhya Pradesh (71%), and among public hospitals (82%), compared with private hospitals (66%). For cumulative odds ordinal logistic regression with proportional odds, having beneficiary's overall satisfaction with the experience of hospitalization under PMJAY as the dependent variable, full likelihood ratio test indicated that the assumption of proportional odds was met, $\chi^2(36) = 46.4$, $p = 0.11$. Likelihood-ratio test indicated that the final model statistically significantly predicted the dependent variable over and above the intercept-only model, $\chi^2(12) = 43.9$, $p < .001$. Table 4 shows that OOP, beneficiary location, and help received at the PM-JAY registration desk have a statistically significant effect on the beneficiary's overall satisfaction with the experience of hospitalization under PMJAY.

**Table 4. Tests of model effects.**

| Independent variable | Wald Chi-Square | Degree of freedom | Level of significance (p-value) |
|---|---|---|---|
| State (Gujrat, MP) | .172 | 1 | .678 |
| Hospital type (Public, Private) | .554 | 1 | .457 |
| OOP made (Yes, No) | 14.568 | 1 | .000 * |
| Beneficiary caste (Marginalized and backward castes, General caste) | .007 | 1 | .935 |
| Beneficiary location (Urban, Non-urban) | 8.251 | 1 | .004 * |
| Beneficiary education (illiterate, primary education, secondary education, graduate and above) | 1.134 | 3 | .769 |
| Type of care/service received (Hospitalization with surgery, Hospitalization without surgery, Daycare procedure) | 1.860 | 2 | .395 |
| Help received at PM-JAY registration desk | 14.110 | 1 | .000 * |
| No of days admitted in hospital | 1.193 | 1 | .275 |

Dependent Variable: Beneficiary's overall experience of hospitalization under PMJAY.

Table 5 shows the Odds ratio for the independent variables of cumulative odds ordinal logistic regression with proportional odds. The odds of beneficiaries who have made OOP having a higher level of satisfaction is 0.24 (95% CI, 0.1–0.5) times that of beneficiaries who have not made OOP, a statistically significant effect, $\chi^2$ (1) = 14.6, p < .0005. The odds of beneficiaries living in an urban area having a higher level of satisfaction is 2.6 (95% CI, 1.4–5.1) times

**Table 5. Odds ratio.**

| Independent variable | | Odds ratio | 95% Wald Confidence Interval for Odds ratio | |
|---|---|---|---|---|
| | | | Lower | Upper |
| **State** | Gujrat | 1.171 | .556 | 2.463 |
| | MP | 1 | . | . |
| **Hospital type** | Private | 1.282 | .666 | 2.469 |
| | Public | 1 | . | . |
| **OOP made** | Yes | .241 | .116 | .500 |
| | No | 1 | . | . |
| **Beneficiary caste** | Marginalized and backward castes | 1.028 | .536 | 1.970 |
| | General caste | 1 | . | . |
| **Beneficiary location** | Urban | 2.623 | 1.358 | 5.066 |
| | Non-urban | 1 | . | . |
| **Beneficiary education** | Illiterate or no formal education | 1.335 | .348 | 5.121 |
| | Primary education | 1.740 | .506 | 5.984 |
| | Secondary education | 1.403 | .389 | 5.053 |
| | Graduate and above | 1 | . | . |
| **Type of care/service received** | Hospitalization with surgery | 1.100 | .440 | 2.751 |
| | Hospitalization without surgery | 1.655 | .673 | 4.068 |
| | Day-care procedure | 1 | . | . |
| **Help received at PM-JAY registration desk** | | 2.004 | 1.394 | 2.880 |
| **No of days admitted in hospital** | | 1.037 | .971 | 1.108 |

Dependent Variable: Beneficiary's overall experience of hospitalization under PMJAY.

that of beneficiaries living in a non-urban area, a statistically significant effect, $\chi^2 (1) = 8.3$, p = .004. An increase in help received at the PM-JAY registration desk is associated with an increase in the odds of beneficiaries having a higher level of satisfaction, with an odds ratio of 2 (95% CI, 1.4–2.9), $\chi^2 (1) = 14.1$, p < 0.0005.

## Information exchange before, during, and after hospitalization

The PM-JAY guidelines indicate that patients should receive information about various processes concerning their hospitalization through calls and SMSs by the scheme authorities. Fig 3 shows that more than half of the patients in Gujarat reported receiving SMS during the verification step; lower proportions of patients reported receiving SMSs for preauthorization, admission, and discharge. In Madhya Pradesh, a significantly lower proportion of patients reported receiving SMSs across processes.

The most commonly reported information exchange was a call after a patient's discharge soliciting their feedback. Nearly half of the beneficiaries in Madhya Pradesh and 60% in Gujarat reported having received a post-discharge feedback call. This proportion was 64% among public hospital beneficiaries of Gujarat and 40% among Madhya Pradesh beneficiaries who received services from public hospitals.

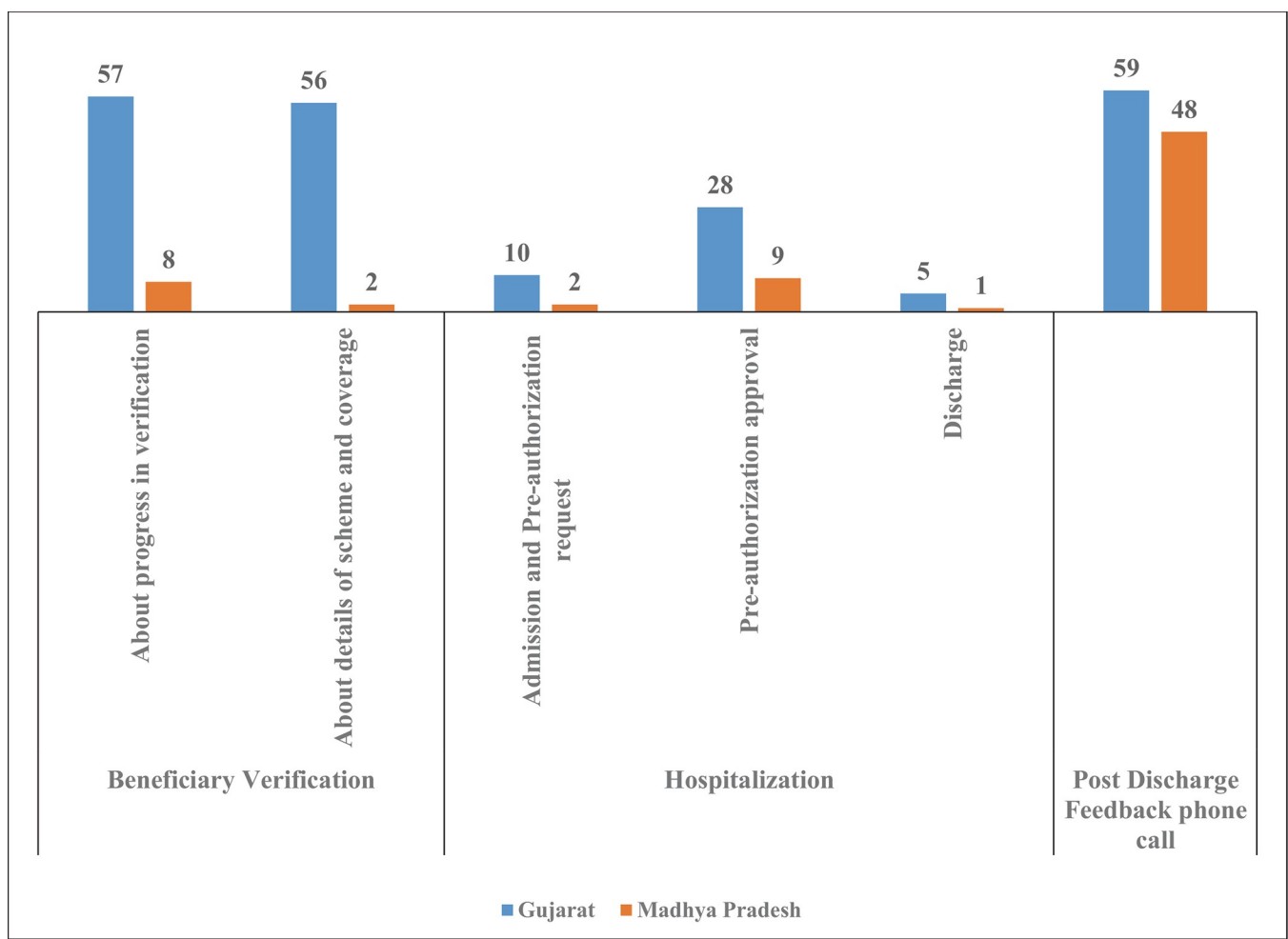

**Fig 3. Proportions of beneficiaries who received SMS/call about specific processes (in %).**

## Discussion

This study aimed to understand beneficiaries' experience of availing healthcare services at the empaneled public and private hospitals in PM-JAY. The results of this study highlight the time delays as experienced by the beneficiaries at different stages of hospitalization and discharge process, the help provided to them at the helpdesk, information exchanged with them by the scheme authorities, financial expense incurred by them, and their satisfaction with hospitalization experience under PM-JAY.

The results of this study highlight that in Gujarat more than 80% of the surveyed beneficiaries were made aware of the scheme and their eligibility for entitlement by the government's official channel (letter from the government, and by the village-level health workers). As compared to it, in Madhya Pradesh most of the beneficiaries got to know about the scheme from informal sources (friends, relatives, newspaper) and on arrival for treatment at a hospital. In the context of Rashtriya Swasthya Bima Yojana (RSBY) an earlier GSHIS in India, only one in four of the beneficiaries knew about the scheme [17]. The extent of usage of GSHISs depends on beneficiaries' level of awareness about the scheme and their entitlements. The absence of such knowledge before the onset of an illness can influence patients' treatment-seeking behavior in terms of them choosing either to delay reaching out to hospitals or deciding not to seek care at all. The lack of targeted awareness generation work, in Madhya Pradesh, through the government's official channels about the scheme and entitlement points to the need for focusing on Information, Education, and Communication (IEC) activities for PM-JAY in the state.

Once the PM-JAY beneficiary reaches a hospital, the PM-JAY helpdesk is supposed to act as a crucial source of information about the scheme, beneficiary's registration and entitlement, and provide support to the beneficiaries in the hospitalization process. In terms of the information and support provided by the helpdesk to the beneficiaries, public hospitals performed much better than private hospitals. In the state of Gujarat, helpdesks in public hospitals were found to be more helpful as compared to that in private hospitals. Helpdesks in public and private hospitals in Madhya Pradesh were not significantly different in terms of providing information and support to the beneficiaries. The results of the regression model highlighted that the help received by the beneficiaries at the PM-JAY helpdesk desk (an indicator of prompt attention) has a significant impact on beneficiaries' satisfaction. In the context of RSBY, an earlier GSHIS in India, the indifference of helpdesks in helping patients and a resultant poor satisfaction among beneficiaries have also been reported [18]. These findings indicate a need to strengthen the functioning of the helpdesk in PM-JAY. This needs to be prioritized for private hospitals as compared to the public hospitals, and for Madhya Pradesh as compared to Gujarat. Such an effort will help in increasing beneficiary's satisfaction and will ultimately have a positive impact on the utilization of the scheme.

Beneficiaries in Gujarat experienced significantly lower waiting time as compared to those in Madhya Pradesh for 'admission and pre-authorization request'–i.e. the time that the AM took in preparing and submitting the request using the PMJAY IT system. Within Gujarat, the waiting time experienced by the beneficiaries for 'admission and pre-authorization request' (an indicator of prompt attention) was not significantly different across public and private hospitals. In the case of Madhya Pradesh, the waiting time experienced by the beneficiaries for 'admission and pre-authorization request' was significantly lower in public hospitals as compared to that in private hospitals. On similar lines, the time taken for 'pre-authorization approval' (an indicator of prompt attention) by the insurer was found to be significantly lower in Gujarat as compared to Madhya Pradesh. The time taken for 'pre-authorization approval' was not significantly different between public and private hospitals in Gujarat. However, for Madhya Pradesh, the time taken for 'pre-authorization approval' was found to be significantly

lower for public hospitals as compared to private hospitals. Time taken for 'pre-authorization approval' can be high if (a) at hospital level AM is submitting incomplete 'admission and pre-authorization request' resulting in repeated queries from the insurer, (b) the insurer lacks the capacity to process the received 'admission and pre-authorization request' on time. It has been suggested that the time required for completing hospital-based processes that involve AMs can be reduced by investing in capacity-building efforts for AMs [15]. The results of the present study highlight a need to focus on capacity-building initiatives for AMs in Madhya Pradesh as compared to Gujarat, and within Madhya Pradesh to prioritize such initiatives for AMs working in private hospitals. The results of this study also highlight a need to investigate insurers' capacity for processing the 'admission and pre-authorization request' on time.

The results of this study highlight that the average length of stay in hospital was not significantly different for surgical and non-surgical treatment. The average length of stay in hospital was found to be significantly lower in Gujarat as compared to Madhya Pradesh. At an aggregate level, the average length of stay was found to be significantly lower in private hospitals as compared to that in public hospitals. Similarly, within Gujarat and Madhya Pradesh, the average length of stay was lower in private hospitals as compared to that in public hospitals. The lower average length of stay in private hospitals as compared to that in public hospitals may be leading to the higher efficiency of hospital bed use with or without having negative consequences on the outcomes of medical treatment. This needs to be investigated in future studies.

The results of this study highlight that around a quarter of the beneficiaries made OOP payments to the hospital. The incidence of OOP payment was high in Madhya Pradesh as compared to Gujarat, and in private hospitals as compared to the public hospitals. The magnitude of the OOP payment was significantly lower in Gujarat as compared to Madhya Pradesh and in public hospitals as compared to private hospitals. The magnitude of OOP payment made was significantly higher for beneficiaries belonging to the marginalized and backward castes as compared to those from general caste. The results of the regression model highlighted that making OOP payments while availing healthcare service under PM-JAY has a significant negative impact on beneficiaries' satisfaction. The high incidence and extent of OOP payments, especially in Madhya Pradesh, is similar to that reported in the context of other GSHIS in India [19–21]. The magnitude of OOP across the two states and type of hospitals as reported in the present study are comparable to the similar estimates among non-insured patients from the recent National Sample Survey [22]. The findings of the present study are also similar to those reported for PM-JAY in the Chhattisgarh state of India. For the Chhattisgarh state of India, it has been reported that PM-JAY neither resulted in a reduction in OOP payment by beneficiaries nor incidences of catastrophic health expenditure, and the magnitude of OOP payment by PM-JAY beneficiaries was significantly higher in private hospitals as compared to that in the public hospitals [23]. The results on OOP of the present study are also consistent with findings from similar schemes in other low- and middle-income countries, where such payments have been stated as a result of moral hazards among poorly-regulated private providers [24].

The OOP payment in PM-JAY has been suggested to be an outcome of the design of the scheme in terms of process flows, poor package rates, and the policy of reserving some procedures for public hospitals. It has been reported that hospitals ask for payments from patients for pre-operative diagnostic procedures due to concern that they will not receive the payment if the patient is not ultimately hospitalized. It has also been reported that patients in private hospitals are asked to make payments for certain multimodal treatments that are reserved for public hospitals. This was found to be more relevant for patients with conditions, like cancer, that require treatment by a team of doctors using multiple treatment modalities. It has also been reported that the patients are asked to pay for the difference of costs between the

treatment preferred by the healthcare providers (for example, medicine or an implant for treatment of a cardiovascular condition) and that covered under the scheme [15]. Irrespective of the reasons, OOP payments affect the financial and physical coverage of the GSHIS scheme and negatively affect user satisfaction. If not addressed optimally, such payments can lead to a paradoxical situation in which an overall increase in the utilization of services results in poor financial and physical coverage [25]. Hence, the results of this study highlight an urgent need for actions aimed at reducing the OOP payment by PM-JAY beneficiaries. The results of this study also highlight a need to understand the reasons for higher OOP payment made by the less-advantaged castes of the society.

In PM-JAY, information exchange with patients (an important element of responsiveness) is based on the timely provision of SMSs and telephone calls from scheme authorities. The results of this study found that this information exchange was better in Gujarat as compared to that in Madhya Pradesh. However, the results also suggest that in both the states this information exchange mechanism is performing sub-optimally as compared to that envisioned in the scheme. This weak information exchange mechanism may be making it easier for hospitals to ask for OOP payment from the beneficiaries, and at the same time making it difficult for beneficiaries to report their grievances to the scheme authorities. This finding, hence, suggests a need to strengthen the information exchange mechanism between patients and scheme authorities, and use of patients' grievances to improve the scheme.

The results of the regression model highlight that beneficiaries' location (rural, urban) has a significant impact on beneficiaries' satisfaction with the experience of hospitalization under PM-JAY. The results highlight that chances of beneficiaries living in urban areas having a higher level of satisfaction were significantly higher than those living in rural areas. This can be because of the long distances that the beneficiaries from rural areas are generally required to travel for reaching hospitals of their choice or due to the issues in services received by them from the hospitals located nearby to them. Moving further, there is a need for PM-JAY scheme authorities to understand the challenges that are faced by the beneficiaries from rural areas and strengthen the scheme to meet their needs.

This study surveyed 200 beneficiaries across Gujarat and Madhya Pradesh. These beneficiaries were selected from the most popular and least popular clusters of each of the 14 hospitals covered in this study. This way of focusing on the most and least popular clusters for each hospital may have introduced some bias in sample selection. Also, for this study, the hospitals to shortlist PM-JAY beneficiaries were selected by the respective SHAs. This may also have unknowingly introduced some biases. In the future, studies may look at alternate ways of sampling the hospitals and patients. Also, the sample size of this study is small. In the future, studies may focus on larger sample size. This study looked at OOP payments made by the beneficiaries to the hospitals. In the future, studies may expand this to include direct and indirect OOP payments made by the beneficiaries in traveling to reach a hospital, making stay arrangements, and forgone wages. This will help in getting a holistic picture of the financial burden faced by the beneficiaries.

## Conclusion

This research contributes to the existing literature by capturing beneficiaries' experience of availing healthcare services at the public and private hospitals empaneled in PM-JAY. The results of this study provide insights into the a) responsiveness of PM-JAY in terms of prompt attention in service delivery, and access to information by the beneficiaries, b) financial burden faced by the beneficiaries in terms of incidence and magnitude of out-of-pocket payments made to the hospital, and c) beneficiary's satisfaction with the experience of hospitalization

under PMJAY and its determinants. These results provide insights to the policy-makers and program managers for strengthening the scheme.

In India, health is a state government subject. For implementing PM-JAY, states have the flexibility to implement the scheme as deemed suitable by them. Before PM-JAY, Gujarat had experience of implementing state-level GSHIS (Mukhyamantri Amrutam scheme) whereas Madhya Pradesh had no such experience. While implementing PM-JAY, Gujarat seems to be leveraging its experience, and in the absence of any such prior experience, Madhya Pradesh seems to be at an experimenting and learning stage. This difference in the capacities of these two governments to conceptualize and implement the PM-JAY scheme is resulting in a significant difference in the experiences of beneficiaries in the two states. To strengthen the implementation of PM-JAY, there is a need for a platform where states can share their experiences and learn from each other. This will help in improving the capacities of the state governments for better implementation of PM-JAY. Along with it, there is also a need for developing a quality assurance mechanism to ensure consistency of outcomes across states.

On the part of program managers, there is a need to look for ways to enhance the responsiveness of the scheme. Prompt attention in service delivery will improve beneficiaries' satisfaction, and timely access to information will enable the beneficiaries to make informed choices and seek accountability from the healthcare providers. For policy-makers, there is an urgent need to focus on steps needed to reduce OOP payment.

## Supporting information

**S1 File. Patient survey questionnaire.**
(DOCX)

**S1 Table. Comparison of study states across selected expenditure parameters.**
(DOCX)

## Acknowledgments

We are thankful to the National Health Authority, India, for the guidance and necessary approvals for the research. We acknowledge the inputs of WHO India team members Dr. Vinod Verma in conducting the research, and Dr. Grace Kabaniha in improving the manuscript. We would also like to thank Dr. Chandrakant Lahariya and Dr. Hilde De Graeve for their support and guidance.

## Author Contributions

**Conceptualization:** Mayur Trivedi, Anurag Saxena, Zubin Shroff.

**Data curation:** Mayur Trivedi, Anurag Saxena, Manas Sharma.

**Formal analysis:** Mayur Trivedi, Anurag Saxena, Manas Sharma.

**Funding acquisition:** Mayur Trivedi, Anurag Saxena.

**Investigation:** Mayur Trivedi, Anurag Saxena.

**Methodology:** Mayur Trivedi, Anurag Saxena, Zubin Shroff.

**Project administration:** Mayur Trivedi, Anurag Saxena, Manas Sharma.

**Software:** Mayur Trivedi, Anurag Saxena.

**Supervision:** Mayur Trivedi, Anurag Saxena, Manas Sharma.

**Validation:** Mayur Trivedi, Anurag Saxena, Zubin Shroff.

**Visualization:** Zubin Shroff.

**Writing – original draft:** Mayur Trivedi, Anurag Saxena.

**Writing – review & editing:** Mayur Trivedi, Anurag Saxena, Zubin Shroff, Manas Sharma.

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
