## [Decision Letter · Decision Letter 0]

1 Apr 2021

PONE-D-21-06856

Improving Responsiveness and Financial Protection in Government-Funded Health Insurance Scheme in India: Evidence from Early Implementation of Pradhan Mantri Jan Aarogya Yojana (AB – PMJAY)

PLOS ONE

Dear Dr. Trivedi,

Thank you for submitting your manuscript to PLOS ONE. After careful consideration, we feel that it has merit but does not fully meet PLOS ONE’s publication criteria as it currently stands. Therefore, we invite you to submit a revised version of the manuscript that addresses the points raised during the review process.

The major comment from all the reviewers is requirement of a more succinct and structured presentation of the manuscript. It requires results and discussion to clearly follow from the objectives of the study. The variables/ indicators need to be clearly defined for responsiveness, quality and financial protection in the methodology section. My own specific comments are added below and also in the attached manuscript. 

The grammar, references and abbreviations must clearly follow the PLOS One style of writing.

We look forward to receiving your revised manuscript.

Kind regards,

Charu C Garg, Ph.D.

Academic Editor

PLOS ONE

Additional Editor Comments:

Studying responsiveness, quality and financial protection for the newly launched PMJAY scheme will add value to Indian policy and is useful for international literature to know factors that influence these attributes for government insurance schemes.

However, the paper needs to be better structured to follow it more clearly, especially for international audience.

There are several Grammatical issues and certain sections are really lengthy. The discussion and conclusion section needs to be better linked to results. The indicators under responsiveness, quality and financial protection must be explained upfront. The abstract needs to be rewritten with clear results and conclusions.

Abstract needs revision. Does not present any results from the text and conclusion does not follow from the result.

The title does not have quality, but you do mention about quality in results. if it is part of responsiveness, please say it clearly. Also AB in the title is not explained or used throughout the manuscript.

Very long introduction, needs to be structured with background linked to objectives studied.

In the methods section, clearly explain the variables/ indicators used for responsiveness, quality and financial protection used in your study. The indicators are mixed up under the procedures - from admission to discharge to follow up. EG. section c has a mix of responsiveness and financial protection in results. Use the same format then for results and discussion. These are given but the presentation needs to be clearer for the reader to follow.

There are several issues in the result section, which have been put as comments in the paper and need to be explained.

Discussion needs to be arranged in the context of major sections studied, what your research says and how does it corroborate with the research from other government health sponsored schemes from past in INDIA such as RSBY or state insurance schemes or other countries. In what ways PMJAY is better than existing insurance schemes and what needs to be improved.

Please see more specific comments in the attached document.

Journal Requirements:

3. Please provide additional details regarding participant consent. In the ethics statement in the Methods and online submission information, please ensure that you have specified how verbal consent was documented and witnessed.

6. We noted in your submission details that a portion of your manuscript may have been presented or published elsewhere. [the contents of this manuscript have not been submitted for publication elsewhere. A modified version of this manuscript is available as a working paper on https://pmjay.gov.in/sites/default/files/2020-02/WP_IIPH_study_2.pdf.] Please clarify whether this publication was peer-reviewed and formally published. If this work was previously peer-reviewed and published, in the cover letter please provide the reason that this work does not constitute dual publication and should be included in the current manuscript.

Reviewers' comments:

Reviewer's Responses to Questions

**Comments to the Author**

1. Is the manuscript technically sound, and do the data support the conclusions?

Reviewer #1: Partly

Reviewer #2: Yes

2. Has the statistical analysis been performed appropriately and rigorously? 

Reviewer #1: No

Reviewer #2: Yes

3. Have the authors made all data underlying the findings in their manuscript fully available?

Reviewer #1: Yes

Reviewer #2: No

4. Is the manuscript presented in an intelligible fashion and written in standard English?

Reviewer #1: No

Reviewer #2: No

5. Review Comments to the Author

Reviewer #1: General comments:

The manuscript reviews major language editing and rewriting. There are far too many language errors across the manuscript, which makes it non readable.

All sections need major editing and shortening. The repetition should be eliminated. The discussion section needs to be improved with editing and in the quality of analysis.

Specific comments:

Abstract need to be completely re-written. It does not capture the essence of manuscript. There are 200 study participants from whom data was collected. Some quantitative analysis findings should be part of the abstract, which currently focuses upon qualitative aspects only. The conclusion in abstract is generic. Please re-write. The acronyms can be avoided.

Financial disclosure: Please confirm if this is full disclosure of grant or was there any other funding in addition to Health policy and system research at WHO HQ? Please also confirm if authors have acknowledged all the key people who contributed to this work?

Ethical statement: was there any specific reason that only verbal consent was sought and not the written consent?

Introduction / main text: The language need major re-writing for academic standards. The wordings such as ; line 3: ‘people may remain unhappy with the health systems” could be avoided. This section is a bit superfluous and verbose. It need to edited and shortened. There are total 108 lines, it is difficult to follow and read. The information about PM-AY can be moved into a box.

Methods section, there is duplication of information and repetitions.

Results section. One of the limitations is that the small sample size and then then findings have been used in both tables and text.

In discussion, the key actionable suggestions should be included. The limitations of this work need to be elaborated. More India specific literature review and citations should be used.

Conclusion: Should be succinct and single or maximum two paragraphs.

Figure 1- 4 are repetition and can be removed.

Reviewer #2: while the theme chosen is good, I have the following observations:

1. The paper may give some context of health care utilization and expenditure about the states chosen

2.Need to strengthen the discussion section with some comparison of the findings of PM-JAY with private health insurance schemes to judge the strengths and shortcomings of PM-JAY.

3. Quantitative analysis may be augmented by estimating odds-ratios for the chosen indicators.

4. The word "improving" may be dropped from the title.

5. There are many grammatical errors and many sentences need editing to have a more terse presentation.

6. in line 251, the figures in the Table and in the description do not match.

7. Line 274, and other lines, the style of writing figures is not uniform.

6. PLOS authors have the option to publish the peer review history of their article (what does this mean?). If published, this will include your full peer review and any attached files.

Reviewer #1: **Yes: **Chandrakant Lahariya

Reviewer #2: No

---

## [Author Response · Author response to Decision Letter 0]

29 Jun 2021

Reviewers' #1 comments:

General comments: The manuscript reviews major language editing and rewriting. There are far too many language errors across the manuscript, which makes it non readable.

All sections need major editing and shortening. The repetition should be eliminated. The discussion section needs to be improved with editing and in the quality of analysis.

Response: We have revised the manuscript substantially with the help of a professional editor. This has reduced grammatical errors and improved the content.

Specific comment 1: Abstract need to be completely re-written. It does not capture the essence of manuscript. There are 200 study participants from whom data was collected. Some quantitative analysis findings should be part of the abstract, which currently focuses upon qualitative aspects only. The conclusion in abstract is generic. Please re-write. The acronyms can be avoided.

Response: We have revised the abstract to reflect these details. 

Specific comment 2: Financial disclosure: Please confirm if this is full disclosure of grant or was there any other funding in addition to Health policy and system research at WHO HQ? Please also confirm if authors have acknowledged all the key people who contributed to this work?

Response: We have revised the financial disclosure statement. It now reads, “The research was made possible through the financial support provided by the World Health Organization. This included support from the Alliance for Health Policy and Systems Research, WHO Geneva as well as from the WHO Country Office, India under grant numbers 68345 (WHO, Geneva) and 67378 (WHO, India). Support was provided under the research programme titled- Health Systems Research for PM-JAY: Improving hospital-based processes for effective implementation. The grant was received by Prof. Dileep Mavalankar, Prof. Mayur Trivedi, and Prof. Anurag Saxena of the Indian Institute of Public Health Gandhinagar. The grant period was from 4 March to 3 September 2019. The URL of the funder is www.who.int. Dr Zubin Shroff of WHO provided guidance in strengthening the methodology and finalizing the instruments, and reviewed the manuscript critically.” 

We have revised the acknowledgment statement. It now reads, “We are thankful to the National Health Authority, India, for the guidance and necessary approvals for the research. We acknowledge the inputs of WHO India team members Dr. Vinod Verma in conducting the research, and Dr. Grace Kabaniha in improving the manuscript. We would also like to thank Dr. Chandrakant Lahariya and Dr. Hilde De Graeve for their support and guidance. Please refer to line number 472-476.

Specific comment 3: Ethical statement: was there any specific reason that only verbal consent was sought and not the written consent?

Response: The respondents as poor beneficiaries of the PMJAY scheme were expected to have a low level of education to be poor and would be illiterate or semi-literate individuals from urban and rural areas. Since the data collection for the survey was done using handheld digital devices, the thumb impression was difficult to record. Therefore, informal verbal consent was obtained. However, the respondents were provided all details of the research, including their rights to refuse to participate and not answer any questions. We have provided these details in the manuscript. Please refer to line number 188-192.

Specific comment 4: Introduction / main text: The language need major re-writing for academic standards. The wordings such as; line 3: ‘people may remain unhappy with the health systems” could be avoided. This section is a bit superfluous and verbose. It need to edited and shortened. There are total 108 lines, it is difficult to follow and read. The information about PM-AY can be moved into a box.

Response: We have revised the introduction substantially. The introduction is also shortened by more than fifteen lines. The line “people may remain unhappy with the health systems” is also removed. The information on PMJAY Is now reduced and thus, not moved in a box. 

Specific comment 5: Methods section, there is duplication of information and repetitions.

Response: We have revised the methodology section substantially to remove duplication of information and repetitions. 

Specific comment 6: Results section. One of the limitations is that the small sample size and then then findings have been used in both tables and text.

Response: We have revised the results section to remove repetitions in the findings that have been used in both tables and text. We have acknowledged the small sample size as one of the limitations. Please refer to line number 438-443.

Specific comment 7: In discussion, the key actionable suggestions should be included. The limitations of this work need to be elaborated. More India specific literature review and citations should be used.

Response: We have revised the discussion section substantially. Certain key actionable suggestions were provided under the conclusion section. These are now shifted to the discussion section. Also, we have added India-specific literature while comparing our findings of awareness of schemes. Please refer to line number 379-381. We have also added literature on waiting time and satisfaction. Please refer to line number 397-398. Additionally, a comparison around OOP has already existed in the manuscript. Please refer to line number 404-406. The limitations of this work are added in the discussion section. Please refer to line number 438-443.

Specific comment 8: Conclusion: Should be succinct and single or maximum of two paragraphs.

Response: We have revised the conclusion section substantially. After shifting certain key actionable suggestions to the discussion section, the conclusion section is now of two paragraphs only. 

Specific comment 9: Figure 1- 4 are repetition and can be removed.

Response: Agreeing to the suggestion, we have removed these figures, as the details are provided in the text. These are now provided as supplementary material.

 

Reviewer #2 comments: 

Specific comment 1. The paper may give some context of health care utilization and expenditure about the states chosen

Response: We have now added this information, along with a Table. Please refer to line number 148-49 and Table 1 for these details. 

Specific comment 2.Need to strengthen the discussion section with some comparison of the findings of PM-JAY with private health insurance schemes to judge the strengths and shortcomings of PM-JAY.

Response: We couldn’t find relevant research on responsiveness and financial protection from the perspective of beneficiaries of private health insurance schemes. Thus, we could not add such a comparison of PMJAY with private health insurance schemes. However, we have compared our findings with other GSHIS. Please refer to line number 406-420 for these details.

Specific comment 3. Quantitative analysis may be augmented by estimating odds-ratios for the chosen indicators.

Response: The study did not analyze the associations or determinants of responsiveness (measured in terms of patient satisfaction) and financial protection (measured in terms of incidence and magnitude of OOP). Estimation of odds ratio was therefore not attempted as it was not related to the study objectives. Additionally, small sample size was a limitation of the study restricting the scope of statistical analysis. We, however, explored the subgroup difference in responsiveness and financial protection across socio-economic categories of the respondents. These details are presented in Table 1 as supplementary material. 

Specific comment 4. The word "improving" may be dropped from the title.

Response: Agreeing to the suggestion, we have removed the word ‘improving’ from the title. 

Specific comment 5. There are many grammatical errors and many sentences need editing to have a more terse presentation.

Response: We have revised the manuscript substantially with the help of a professional editor. This has reduced grammatical errors and improved the content.

Specific comment 6. In line 251, the figures in the Table and in the description do not match.

Response: These were because the findings were presented to highlight the proportion of the respondents who did not make any OOP payments. We have now revised the language and figures to reflect the information in terms of those who made the OOP payment, as presented in the table

Specific comment 7. Line 274, and other lines, the style of writing figures is not uniform.

Response: We have corrected the style of writing figures to make them uniform at all places. 

 

Specific comments in the attached document:

Comment 1: check grammar. 

Response: We have revised the abstract and edited it for grammatical errors. 

Comment 2: The abbreviation in title and text should be the same, otherwise need to repeat the full form here as well.

Response: We have revised the abbreviation in the title. 

Comment 3: Indicate dollar values also

Response: We have revised the abstract and added dollar values. 

Comment 4: delete. Repeated

Response: We have revised the abstract accordingly.

Comment 5: Avoid abbreviation 

Response: We have revised the abstract accordingly.

Comment 6: The conclusion seems more like objective and does not follow from results. 

Response: We have revised the conclusion part of the abstract accordingly.

Comment 7: Very long introduction. Clearly provide the background in the context of the objective of the study.

Response: We have revised the introduction substantially. The introduction is also shortened by more than fifteen lines.

Comment 8: Please place references as per the plos one style - different places before and after full stops.

Response: We have revised the references as per PLOS style. They are now after full stops at all places. 

Comment 9: check grammar. 

Response: We have revised the entire document for grammatical errors. 

Comment 10: Major features to be more succinct and summarized in a box under headings

Response: We have revised the introduction of PM-JAY substantially and now it is shorter, and in the flow of the text. At this stage, it need not be in the box. However, if the reviewer feels appropriate, we would be happy to put the details in the box. 

Comment 11: Please provide full references with links and dates accessed as per PLOS style.

Response: We have revised this reference. We have used the ‘PLOS’ style in the EndNote referencing software for generating full references for this manuscript. 

Comment 12: what are autonomous hospitals

Response: These are publicly funded hospitals that have special status through legislation and thus, are governed independently. However, since this was not relevant for the objectives, we have removed this sentence. 

Comment 13: what if the claim os from the same episode of ailment.

Response: All patients with single claims were retained in the population list. Few patients required multiple sessions of treatment like dialysis and chemotherapy. In such cases, each session was claimed separately, and thus, there were multiple claims of those patients. We retained all these patients. 

Comment 14: Would this not add a bias for responsiveness, as the hospitals with least claims can also be the ones where people did not want to go.

Response: The hospitals for the study were selected by the state authorities. It was important to ensure the representation of patients across hospitals and specialties within the hospitals. The use of multi-stage sampling method with the use of Probability Proportional to Size ensured that the selection of respondents was not biased either in favour or against any type of specialty and hospital. This information is now explained in the manuscript. Please refer to line number 169-171.

Comment 14: They would already have better knowledge of the health scheme, so their responsiveness may be better.

Response: The selection of the state with previous experience of implementing a GSHIS was an element of study at the time of selection of the study location. The responsiveness in terms of their registration in the scheme was not analyzed and therefore, ‘n’ for the beneficiary registration did not include these patients. The responsiveness of their hospitalization experience was measured only for the particular episode during the study duration. Thus, such possibility of their experience influencing the knowledge and responsiveness of the scheme may be ruled out. 

Comment 15: Can be added as supplementary material

Response: We have submitted a copy of the questionnaire as supplementary material. 

Comment 15: Would this not create seasonal bias

Response: The study population includes patients who availed hospitalization under PMJAY during April 2019. This was given by the state authorities, and thus, was not under the control of the researchers. The data collection was done during May-June 2019. There could be seasonal effects on the type of illnesses, but the responsiveness and financial protection may not get affected by the time of the year. 

Comment 16: Can say informed consent was taken

Response: We have revised the statement and provided more details in the manuscript. Please refer to line number 190-194.

Comment 17: These can be summarized under different headings in a box - in the same format as the results are presented. Some of it also repeated from above. Need to add them together.

Response: These are now well-defined and discussed in the methodology section between lines 196-211. We have removed the repetition. The results of this study were documented using the flow of processes in the standard official guideline as a template. The findings are therefore divided into subsections that correspond to the PM-JAY guidelines for hospital-based transactions. These sub-sections are beneficiary identification and registration, treatment package selection and blocking, discharge and payments, and information exchange.

Comment 18: Is anything covered before and after hospitalization. Was information collected on experience before and after hospitalization? If referral etc. had the influence on registration process. What about coverage after hospitalization.

Response: The study collected information on pre-and post-hospitalization expenses in detail. The analysis of OOP payment includes these details in light of the provision of such coverage under the scheme. The information exchange is an in-built element regarding a) registration done before or at the time of the hospitalization, b) package selection during hospitalization, and c) calls for feedback after the hospitalization. These were studied in detail.

Comment 19: why is education of beneficiary/ or highest education in HH not taken as indicator, as awareness and responsiveness depends quite a bit on that.

Response: We have now added this information in Table 3.

Comment 20: mismatch - 61 vs 74. Also if they were already enrolled in a scheme, one would expect better experience about the process. Could it be they did not want to switch.

Response: This is not a mismatch. 74% of the total 61 respondents (i.e. 45 respondents) indicated that they knew about the scheme from the letter. The remaining 39 participants at the time of the research were yet to be transferred from the MA scheme to PMJAY. For all practical purposes, all beneficiaries, irrespective of the scheme received the same coverage for their hospitalization. We did not include these 39 participants for their registration experience as they were not registered in PMJAY till April 2019. The participants did not have choice in switching as the existing scheme was getting subsumed under PMJAY over time. 

Comment 21: what was the difference in public vs. private will be useful to know

Response: The satisfaction was higher among beneficiaries from public sector hospitals (83%) than their counterparts from the private sector (77%). This is now added in the result section. Please refer to line number 296 and 297.

Comment 22: You mention preauthorization above also. These are longer than for admission and preauthorization. Do you mean only admission above? Otherwise hours should be more above.

Response: The first mention of pre-authorization is in the context of ‘admission and pre-authorization request’ – i.e. time that the AM took in preparing and submitting the request. The second mention is about ‘pre-authorization approval’ to be received from the scheme authorities. The first one is the efficiency of the insurance processes at the level of healthcare providers, and the latter is about the efficiency of the scheme managers at the level of the insurance company / Trust. These details are clarified and streamlined in the subsection of ‘treatment package selection and blocking’ in the result section. 

Comment 23: Why is the sum of total smaller than public plus private

Response: The total of ‘n’ is 100 (after adding 56 private and 44 public). The incidence of OOP is in proportion. So, the total need not be the summation of the two proportions. It is the mean of the two (i.e. 13 and 9). 

Comment 24: Does PMJAY not cover the medicine bills for medicines not available. Also for a procedure, would everything not be covered? 

Response: Ideally, all expenses should be covered. According to the guideline, the patient need not pay and the hospital must not do balance billing. However, the finding of this study, therefore, brings evidence on OOP and connects it to the reasons for such OOP in the discussion section. We, therefore, raise the need to address underlying supply-side reasons for OOP.

Comment 25: Are they informed and consulted about it before the registration.

Response: Yes. In most cases, the patients were informed beforehand about the possibility of such part payments. This was ascertained during the survey through informal discussions with the respondents. 

Comment 26: Is insurance agency or the scheme informed about these payments charged to the patients?

Response: The survey respondents did not know of such communication. Therefore, this is not part of our analysis. However, since part payment is not allowed under the scheme, there is no mechanism/ mandate for the hospitals to make this information available to the insurer/scheme managers. 

Comment 27: This is really high. Can this be compared with OOPS data from HH survey.

Response: This is high in the case of patients who availed of hospitalization in a private hospital in Madhya Pradesh, a state which saw such GSHIS with 5 lac coverage for the first time. Therefore, this needs to be seen contextually for the GSHISs covering secondary and tertiary hospitalization. We refrained from comparing the magnitude of OOP of this study (which was measured for those who made some payment) with similar data from the HH survey as the study population is not comparable. Our study population comprises those who exclusively opted for secondary and tertiary hospitalization, and thus are not comparable with households incurring expenses on a range of healthcare, including primary care and outpatient treatment. We have compared the incidence of OOP among the insured patients with similar research on beneficiaries of past GSHIS like RSBY that offered relatively low coverage. Please refer to line number 406-420 for these details.

Comment 28: This should and most of it is part of introduction. Not required here.

Response: We have revised this paragraph and removed irrelevant sentences from here. 

Comment 29: What about the links to the socio economic conditions of the households. Also does education matter. 

Response: The study did not analyze the determinants of responsiveness (measured in terms of patient satisfaction) and financial protection (measured in terms of incidence and magnitude of OOP). However, on comparing these parameters across the socio-economic subgroups of the respondents, it was found that the satisfaction for registration (91%) and hospitalization (82%) was highest among the illiterate respondents. This sub-group has the lowest incidence of OOP (15%), but reported the highest amount of OOP payment (Rs. 41075). We have provided this information with a table as supplementary material. 

Comment 30: What about the links to the rights and information. Also not discussed clearly about the payments and links to Aarogya Mitra. Should AM not play a role in helping the patients to keep their payments low. Even if AM desk is not there in hospitals, should AM not visit the hospitalized patient under the scheme. That would improve the quality and financial protection. 

Response: Aarogya Mitra is a staff of the hospital appointed to streamline the liaison between patients, doctors, and scheme authorities. As an employee of the hospital, the AM interacts with the patients on behalf of the hospitals – at the helpdesk or bedside. The AM’s interaction with the patient is limited to information and documentation for preauthorization and discharge. As for the payment, the AM merely conveys the decision of ‘balance payment’, if any, taken by the hospital authorities. Lastly, being a household survey of the patients, this research did not dwell deeper in the supply side issues of how AM can play an improved role at the provider level. 

Comment 31: how do this and some other conclusions follow from your results. Please restrict discussion and conclusions to what your results show.

Response: This is connected to the reasons that patients indicated for their OOP, especially in the case of those who made a lump sum payment. These patients reported that they were informed at the helpdesk that the actual costs of their treatments or procedures were higher than the amount the hospital would receive from claim reimbursements from PM-JAY; thus, patients needed to pay the balance. This is indicated in the result section (Please refer to line number 32-335). We have revised the language and added corresponding arguments and evidence (Please refer to line number 414-416).

---

## [Decision Letter · Decision Letter 1]

4 Sep 2021

PONE-D-21-06856R1Responsiveness and Financial Protection in a Government-Funded Health Insurance Scheme: Evidence from Early Implementation of Pradhan Mantri Jan Aarogya Yojana (PM-JAY) in IndiaPLOS ONE

Dear Dr. Trivedi,

Thank you for submitting your manuscript to PLOS ONE. After careful consideration, we feel that it has merit but does not fully meet PLOS ONE’s publication criteria as it currently stands. Therefore, we invite you to submit a revised version of the manuscript that addresses the points raised during the review process.

The revision has addressed some comments, however the paper still does not meet the standard of the journal in terms of rigorous statistical analysis. Many comments from of the previous reviewers are still not met which have also been raised by the secondary reviewer chosen for this revision. The abstract is still vague with conclusions not following from results. None of the conclusion and seem to cater to the international audience and are very broad even for Indian perspective. Further the discussion section is still very weak and does not clearly discuss  the results from the study in comparison to similar studies internationally or in comparison to the control population – that is those who did not benefit from the PMJAY scheme. 

We look forward to receiving your revised manuscript.

Kind regards,

Charu C Garg, Ph.D.

Academic Editor

PLOS ONE

Additional Editor Comments (if provided):

The title has responsiveness and financial protection. please discuss these clearly.

Abstract is still very vague. The introduction must have the objective linked to the title. Methods must state how are responsiveness and financial protection studied in the context of PMJAY scheme. Results should be linked clearly to the indicators measured and conclusion should be linked to the results.

Conclusion: there are unnecessary sentences in the conclusion. Eg. The statement “This study provides insights from its beneficiaries based on their experiences of hospitalization under PM-JAY”  —this is objective and not conclusion.

Need to clearly say what GSHIS did to improve responsiveness and what was the outcome or the result of that and what can India and other counties learn from the experience. 

Please rewrite the abstract clearly

Introduction: Liner 95 – not clear - 500 million beneficiaries from the 100 million poor and vulnerable families

Last para of introduction could clearly state how responsiveness and financial protection would be studied. Especially there is nothing about financial protection in the introduction.

Table 1: the data should not be a part of the methodology. Even if is based on secondary sources, these are results. However not being a part of the main study, these could be supplementary material and the figures can be used in the context of responsiveness and financial protection in discussion. The hospital OOP data can be found in the latest version of the NSSO key indicators for 2017-18. Plus, the OOP differs when hospitalization is in public or private facilities and in rural and urban areas. The NHA data is also dated and available for more recent years

Methods: Still; very weak to compare the two states. No clear statistical analysis to understand the differences in the two states. Just presenting table 1 in methods does not show how the results are compared across two states and why the differences are observed. It’s just data presented from a survey with no good analysis. Can the results be compared with control population, those who had similar hospitalization episodes but were not a part of PMJAY scheme. 

Results: Why are the n in tables 4 and 5 different. Tables 4 and 5 can be combined.

There are no control variables to say whether the PMJAY was more responsive or not.

Discussion: Lines 370-371 add OOPE also.

The discussion should be clearly organized first around the variables for responsiveness providing quantitative comparison with the studies from India and international. Why do you say PMJAY is more responsive – in comparison to what.

Also then compare financial protection the same way.

Lines 383, can you pls compare with the references mentioned in quantitative terms to provide a perspective of improvement. In providing the information to patients.

Lines 389-90: Do you mean overall in both states public sector hospitals performed better.

Discussions should clearly bring out reasons for differences in waiting times at the public and private sector and reasons for it. Also compare quantitatively with other studies to show whether the waiting time mentioned is responsive. Line 401- you mention short waiting time- but it was 3 hours in private sector in MP. That is not short.

Lines 406-420. OOP discussion should again be in perspective with non-insured patients and also insured in other programs. Why is PMJAY better. The para provides a weak discussion.

There are still a lot of repetition in conclusions and needs to be a para highlighting important points.

Is there a rural urban variation in responsiveness and financial protection? Why are there such varied results between two states. What is working in one state and not the other? The paper is important but still not convincing enough. 

Reviewers' comments:

Reviewer's Responses to Questions

**Comments to the Author**

1. If the authors have adequately addressed your comments raised in a previous round of review and you feel that this manuscript is now acceptable for publication, you may indicate that here to bypass the “Comments to the Author” section, enter your conflict of interest statement in the “Confidential to Editor” section, and submit your "Accept" recommendation.

Reviewer #2: All comments have been addressed

Reviewer #3: (No Response)

2. Is the manuscript technically sound, and do the data support the conclusions?

Reviewer #2: Yes

Reviewer #3: Partly

3. Has the statistical analysis been performed appropriately and rigorously? 

Reviewer #2: Yes

Reviewer #3: No

4. Have the authors made all data underlying the findings in their manuscript fully available?

Reviewer #2: Yes

Reviewer #3: No

5. Is the manuscript presented in an intelligible fashion and written in standard English?

Reviewer #2: Yes

Reviewer #3: No

6. Review Comments to the Author

Reviewer #2: The authors have addressed the comments raised in the review, revised the language and grammar and the paper may be accepted now.

Reviewer #3: Thanks for taking up an interesting topic on evaluation of delivery of Government Health Insurance Schemes to poor which was revised and re-launched in Sept 2018. I have reviewed the revised version of the paper and looking at previous reviewers comments I would like to state that all comments are not adequately addressed. My additional comments are as follows. 1. Abstract need a thorough revision which I agree with previous reviewers. 2. Neither objectives nor main outcome measurements were clearly defined in the paper as well as in the abstract. Conclusion seems don't appear from the findings of the survey. 3. No statistical tools were applied to explain differences in two states or controlling for hospital types. 4. It appears sample is biased when we talk about share of SC & ST patients in Madhya Pradesh. 5. The major flaw is in the sample selection procedure where the just did quota sampling from two tales of the patients hospitalised during April 2019 in 14 hospitals. One can't portray a robust picture using top 10% and bottom 10% of patients using services as "Highest"/ "Lowest" consumption of health care resources in the sampled hospitals. Instead of following a randomisation process they eliminated the middle 80% of cases. 6. I have mentioned comments on the body of paper and on tables; e.g. I can't understand they are using "Total" heading in both columns and rows even whilst presenting statistics on average for two states.

7. PLOS authors have the option to publish the peer review history of their article (what does this mean?). If published, this will include your full peer review and any attached files.

Reviewer #2: No

Reviewer #3: **Yes: **Anil Gumber

---

## [Author Response · Author response to Decision Letter 1]

3 Mar 2022

• Comment: The title has responsiveness and financial protection. please discuss these clearly.

Response: The meaning of responsiveness and its dimensions are discussed on line number 72 – 82. The details of responsiveness and financial protection used in the study are discussed from line number 138-144, and indicators of responsiveness and financial protection are detailed in line number 211 – 227. We have revised the title to “Experiences and challenges in accessing hospitalization in a Government-Funded Health Insurance Scheme: Evidence from Early Implementation of Pradhan Mantri Jan Aarogya Yojana (PM-JAY) in India”.

• Comment: Abstract is still very vague. Please rewrite the abstract clearly

Response: We have revised the abstract to have objectives linked to the title of the study, methods stating the way indicators of responsiveness and financial protection were defined, and conclusion following from the results. Please refer line number 27 – 61. 

• Comment: Introduction: Line 95 – not clear - 500 million beneficiaries from the 100 million poor and vulnerable families.

Response: We have revised this. Please refer line 90 – 96. 

• Comment: Introduction could clearly state how responsiveness and financial protection would be studied. 

Response: We have added the details. In line 72-82 the meaning of responsiveness and its dimensions are discussed. Towards the end of the Introduction section, in line 138 -144 further details of the way this study attempts to study responsiveness and financial protection have been specified. In the Methods section, the indicators of responsiveness and financial protection used in the study are discussed from line number 211 – 227.

• Comment: Table 1: the data should not be a part of the methodology. Even if is based on secondary sources, these are results. However not being a part of the main study, these could be supplementary material and the figures can be used in the context of responsiveness and financial protection in discussion. The hospital OOP data can be found in the latest version of the NSSO key indicators for 2017-18. Plus, the OOP differs when hospitalization is in public or private facilities and in rural and urban areas. The NHA data is also dated and available for more recent years

Response: We have updated the table with the latest data. We have also taken the table to supplementary material. Please refer to Supplementary Table 1. 

• Comment: Was data collection completed within one month. 

Response: No. The sample was derived from the patients who availed treatment during this one month. The study duration was March – August 2019. We have worded the description accordingly. Please refer line number 154, and 169. 

• Comment: Methods- Still; very weak to compare the two states. No clear statistical analysis to understand the differences in the two states. Just presenting table 1 in methods does not show how the results are compared across two states and why the differences are observed. It’s just data presented from a survey with no good analysis. The paper lack rigorous statistical analysis. No statistical tools were applied to explain differences in two states or controlling for hospital types. 

Response: We have revised the methods sections. We have used Non-parametric test (Mann–Whitney U test) to compare beneficiaries’ responses across states (Gujarat and Madhya Pradesh), hospitals across states (public and private hospitals), hospitals within a state (public and private hospital in Gujarat, public and private hospital in Madhya Pradesh), and across socio-economic categories. We have also carried out cumulative odds ordinal logistic regression with proportional odds to determine the effect of state, hospital type, OOP incidence, beneficiary’s caste, beneficiary’s location, beneficiary level of education, type of care/services received, help received at PM-JAY registration desk, and the number of days for which the beneficiary was admitted in a hospital (independent variables), on the beneficiary’s satisfaction with the experience of hospitalization under PMJAY (dependent variable). Please refer line numbers 233-272. Please refer to the results section for the results of the Non-parametric test (Mann–Whitney U test) and cumulative odds ordinal logistic regression with proportional odds. Please refer to the results section for the description of results obtained. 

• Comment: Results- Why are the n in tables 4 and 5 different. Tables 4 and 5 can be combined.

Response: Table 4 was about the incidence of OOP and thus ‘n’ refers to those who made such expenses. Table 5 was about the magnitude of OOP payment for those who experienced OOP. We have combined Tables 4 and 5. Please refer to Table 3 on line number 422. 

• Comment: No Differences between two states were tested by each socio-economic conditions

Response: We have used Non-parametric test (Mann–Whitney U test) to test the differences by each socio-economic condition. Please refer line number 445 – 450. We have also used caste, level of education, location of residence as independent variables in cumulative odds ordinal logistic regression with proportional odds. Please refer line 246-272, refer to table 4 and 5 at line 471 and 482. 

• Comment: The discussion section is very weak and does not clearly discuss the results of the study. None of the points seem to cater to the international audience and are very broad even for Indian perspective

Response: We have revised the discussion section to highlight the significance of the results. We have also compared the findings of this study with other studies carried out in national and international contexts. We have also discussed reasons for difference in performance across states and types of hospitals. Please refer to line from 500 – 637. 

• Comment: Lines 383, can you pls compare with the references mentioned in quantitative terms to provide a perspective of improvement in providing the information to patients.

Response: Information provided. Please refer to line 511 – 512. 

• Comment: OOP discussion should be in perspective with non-insured patients and also insured in other programs.

Response: We have updated the discussion section. In the discussion section, results related to OOP payment are discussed in comparison with the OOP payment made by non-insured, those insured under PM-JAY in other states of India. Please refer line number 568 – 588. 

• Comment: Neither objectives nor main outcome measurements were clearly defined in the paper. The conclusion seems don't appear from the findings of the survey.

Response: The objectives and aims of the study, outcome measures are outlined in the introduction section. Please refer line 130 – 144. The details of the way outcome measures were operationalized and data was collected are provided in line number 208-228. We have thoroughly revised the results, discussion, and conclusion sections of the paper. 

• Comment: There is still a lot of repetition in conclusions and there needs to be a para highlighting important points.

Response: We have revised the conclusion part to highlight important takeaways. Please refer to lines 639-663. 

• Comment: It appears sample is biased when we talk about the share of SC & ST patients in Madhya Pradesh. 

Response: The profile of survey respondents as provided in table 2 on line 292 was a finding of the study. The sample of respondents for the study was drawn from the list of claims provided by the state health agency of the respective states. These were patients who received treatment from sample hospitals during April 2019. Scheduled Tribe, Scheduled Caste, Other Backward Caste represent the marginalized section of society in India. Taken together beneficiaries from these castes constitute 74% of the sample of beneficiaries from Madhya Pradesh. 

• Comment: The major flaw is in the sample selection procedure where they just did quota sampling from two tales of the patients hospitalized during April 2019 in 14 hospitals. One can't portray a robust picture using the top 10% and bottom 10% of patients using services as "Highest"/ "Lowest" consumption of health care resources in the sampled hospitals. Instead of following a randomization process, they eliminated the middle 80% of cases. 

Response: The sample of this study did not eliminate 80% of cases. We applied the ‘top-bottom’ rule` to the specialties, and not to patients, at each of the 14 sample hospitals for which we received the claims data from the insurance authorities. The hospitals included in this study were single- and multi-speciality in nature, and thus, we decided to include a representation of patients across a range of services, by selecting clusters having top 10% (most popular services) and bottom 10% loads of patients (Least popular services) from each hospital. These are not 10% patients, but clusters. By following this procedure, we had a pool of 57% of patients (67% in Gujarat and 53% in Madhya Pradesh) from which we randomly selected the respondents. There is a possibility that this way of selecting the respondents may have introduced some bias. We have acknowledged this a possibility in the discussion section of the paper. Please refer to lines 626-633. 

• Comment: I have mentioned comments in the body of the paper and on tables; e.g. I can't understand they are using "Total" heading in both columns and rows even whilst presenting statistics on average for two states.

Response: We have revised the paper to take care of the comments received in the body of paper.

---

## [Decision Letter · Decision Letter 2]

29 Mar 2022

Experiences and challenges in accessing hospitalization in a Government-Funded Health Insurance Scheme: Evidence from Early Implementation of Pradhan Mantri Jan Aarogya Yojana (PM-JAY) in India

PONE-D-21-06856R2

Dear Dr. Trivedi,

We’re pleased to inform you that your manuscript has been judged scientifically suitable for publication and will be formally accepted for publication once it meets all outstanding technical requirements.

Kind regards,

Charu C Garg, Ph.D.

Academic Editor

PLOS ONE

Additional Editor Comments (optional):

As suggested by the reviewer as well, professional English editing can help the paper to be more crisp and bring further clarity. 

Reviewers' comments:

Reviewer's Responses to Questions

**Comments to the Author**

1. If the authors have adequately addressed your comments raised in a previous round of review and you feel that this manuscript is now acceptable for publication, you may indicate that here to bypass the “Comments to the Author” section, enter your conflict of interest statement in the “Confidential to Editor” section, and submit your "Accept" recommendation.

Reviewer #3: All comments have been addressed

2. Is the manuscript technically sound, and do the data support the conclusions?

Reviewer #3: Partly

3. Has the statistical analysis been performed appropriately and rigorously? 

Reviewer #3: Yes

4. Have the authors made all data underlying the findings in their manuscript fully available?

Reviewer #3: No

5. Is the manuscript presented in an intelligible fashion and written in standard English?

Reviewer #3: No

6. Review Comments to the Author

Reviewer #3: Most of my comments are addressed. I think the manuscript still needs English Editing. Tables titles need more clarity.

7. PLOS authors have the option to publish the peer review history of their article (what does this mean?). If published, this will include your full peer review and any attached files.

Reviewer #3: **Yes: **Anil Gumber

---

## [Editor Report · Acceptance letter]

21 Apr 2022

PONE-D-21-06856R2 

Experiences and challenges in accessing hospitalization in a Government-Funded Health Insurance Scheme: Evidence from Early Implementation of Pradhan Mantri Jan Aarogya Yojana (PM-JAY) in India 

Dear Dr. Trivedi:

I'm pleased to inform you that your manuscript has been deemed suitable for publication in PLOS ONE. Congratulations! Your manuscript is now with our production department. 

Kind regards, 

on behalf of

Dr. Charu C Garg 

Academic Editor

PLOS ONE